# Robust Bidirectional Associative Memory via Regularization Inspired by the Subspace Rotation Algorithm

## Abstract

Bidirectional Associative Memory (BAM) trained by Bidirectional Backpropagation (B-BP) suffer from poor robustness and sensitivity to noise and adversarial attacks. To address it, we propose a novel gradient-free training algorithm, the Bidirectional Subspace Rotation Algorithm (B-SRA), designed to improve the robustness and convergence behavior of BAM. Through comprehensive experiments, two key principles, orthogonal weight matrices (OWM) and gradient-pattern alignment (GPA), are identified as central to enhancing the robustness of BAM. Motivated by these insights, new regularization strategies are introduced into B-BP, yielding models with significantly improved resistance to corruption and adversarial perturbations. We conduct an ablation study across different training strategies to determine which approach achieves a more robust BAM. Additionally, we evaluate the robustness of BAM under various attack scenarios and across increasing memory capacities, including the association of 50, 100, and 200 pattern pairs. Among all strategies, the SAME configuration—which combines OWM and GPA—achieves the highest resilience. Our findings suggest that B-SRA and carefully designed regularization strategies lead to more reliable associative memories and open new directions for building resilient neural architectures.

## 1 Introduction

Over the past decades, numerous researchers have sought to enhance the robustness, stability, and retrieval fidelity of associative memory (AM). Early studies investigated noise-tolerant learning rules and stability analyses (Feng & Plamondon, 2003; Leung et al., 1995; Hassoun & Watta, 2020), while others explored improved bidirectional mapping mechanisms and energy-based formulations to reduce retrieval errors (Giles & Maxwell, 1987; Acevedo-Mosqueda et al., 2013). More recent work has examined AM under adversarial and uncertain environments, highlighting vulnerability to noise and perturbations in both theoretical and practical settings (Karunaratne et al., 2021; Zhang & Zeng, 2023). Techniques for enhancing stability, such as regularization, pruning strategies, and biologically inspired learning rules, have also been proposed to improve resilience under challenging conditions (Strock et al., 2020).

Despite these efforts, BAM trained via gradient-based methods such as Bidirectional Backpropagation (B-BP) (Adigun & Kosko, 2019; Kosko, 2021; Rosenblatt et al., 1962) still suffer from several major limitations. These include slow convergence, high sensitivity to initialization and hyperparameters, and vulnerability to noise and adversarial attacks. Such weaknesses hinder the deployment of BAM in real-world applications, particularly those requiring robustness under uncertainty, such as biometric authentication (Zhang & Yang, 2023), autonomous systems (Hsu et al., 2023), and secure communications (Paraiso et al., 2021).

Recent advances in associative memory architectures, such as Dense Associative Memories (DAM) (Krotov & Hopfield, 2016) and Modern Hopfield Networks (MHN) (Ramsauer et al., 2022), offer increased capacity and stability. However, both DAM and MHN function more like feedforward neural networks rather than true BAM when it comes to pattern retrieval. The challenge of achieving robust associative retrieval, particularly under noisy or adversarial conditions, remains unresolved.

## 1.1 MOTIVATION AND CONTRIBUTION

Traditionally, B-BP is a popular algorithm for training BAM (Adigun & Kosko, 2019). Inspired by recent advances in training associative memories through subspace rotation techniques (Lin et al., 2024; 2023; 2025), we extend SRA from RHN to BAM. Specifically, we propose the Bidirectional Subspace Rotation Algorithm (B-SRA), a novel, gradient-free training method that enhances the robustness and convergence speed of BAM by directly optimizing their weight matrices through subspace rotation, thus avoiding the limitations of gradient-based approaches like B-BP. Inspired by BAM trained by B-SRA, we further proposed two regularization that could improve the robustness of BAM significantly. The key contributions of this paper are summarized as follows:

- **Extension of SRA to BAM**: We have proposed the Bidirectional Subspace Rotation Algorithm (B-SRA), a gradient-free training method for BAM, which accelerates convergence and enhances robustness against adversarial attacks.

- **Introduction of Regularization to B-BP**: By analyzing the behavior of BAM trained with B-SRA, we propose two regularizers—Orthogonal Weight Matrix (OWM) and Gradient-Pattern Alignment (GPA)—and incorporate them into B-BP to improve the robustness of BAM.

- **Evaluation of BAM Robustness Against Adversarial Attacks**: The performance of BAM trained with B-SRA, B-BP, and B-BP regularized with OWM and GPA is evaluated under several adversarial attacks, including the Fast Gradient Sign Method (FGSM) (Goodfellow et al., 2014), Fast FGSM (FFGSM) (Wong et al., 2020), Basic Iterative Method (BIM) (Kurakin et al., 2018), Projected Gradient Descent (PGD) (Madry et al., 2017), and Gaussian Noise (GN).

## 1.2 ORGANIZATION

The rest of the paper is organized as follows: In Section 2, we introduce the definition of BAM and describe its dynamical behavior in detail. In Section 3, we discuss the underlying mechanism of the Subspace Rotation Algorithm (SRA) and propose B-SRA for training BAM. Section 4 presents a comprehensive experimental evaluation of BAM trained using different strategies, including B-BP, B-SRA, and B-BP with the proposed regularizers. We analyze robustness under various conditions, such as corrupted inputs, GN, and adversarial attacks (FGSM, FFGSM, BIM, PGD). Finally, in Section 5, we conclude that B-SRA outperforms B-BP in training a robust BAM. Inspired by B-SRA, B-BP With the OWM and GPA regularizations can further enhance the resilience of BAM under various adversarial attacks.

## 2 BIDIRECTIONAL ASSOCIATIVE MEMORY

Assuming we have a BAM with $K$ layers, meaning we have $K$ layers of weight matrix, which are indexed as $W_1, W_2, \cdots, W_K$. The paired patterns are called A and B. Without loss of generality, let the input pattern A be considered as the first hidden layer 0 and the input pattern B as the last hidden layer $K$, so the layers can be indexed as $h_0, h_1, \cdots, h_K$.

In the path from pattern A to pattern B, the signal before activation is represented by $U$, indexed as $U_1, U_2, \cdots, U_K$, and the signal after activation is represented by $H$, indexed as $H_0, H_1, \cdots, H_K$. Note that $H_0$ is actually the input pattern A. In the path from pattern B back to pattern A, the reconstructed signal before activation is represented by $R$, indexed as $R_K, \cdots, R_0$, corresponding to $U_1, U_2, \cdots, U_K$, and the signal after activation is represented by $V$, indexed as $V_K, V_{K-1}, \cdots, V_1$, corresponding to $H_0, H_1, \cdots, H_K$.

The dynamical behavior of the BAM can then be described as follows:

**In the Path from A End to B End:**

$$\frac{dU_k(t)}{dt} = W_k H_{k-1}(t), \quad H_k(t) = g \odot (U_k(t)), \quad k = 1, 2, \ldots, K. \tag{1}$$

where $U_k(t)$ is the pre-activation state of layer $k$, and $H_k(t)$ is the post-activation state computed using a non-linear activation function $g \odot (\cdot)$, which is an element-wise operation and is chosen as tanh in our study. $W_k$ is the weight matrix at layer $K$. The state $U_k(t)$ evolves dynamically over time as the input $H_{k-1}(t)$ propagates through the network, producing the output $H_k(t)$ for each layer.

**In the Path from B End to A End:**

$$\frac{dR_{k-1}(t)}{dt} = V_k(t)W_k^T, \quad V_{k-1}(t) = g \odot (R_{k-1}(t)), \quad k = K, \dots, 1. \tag{2}$$

where $R_k(t)$ is the pre-activation state of layer $k$, and $V_k(t)$ is the post-activation state computed using $g \odot (\cdot)$, which is also an element-wise operation and is chosen as tanh in our study. $W_k^T$ is the transpose of the weight matrix for backward signals at layer $k$. In this context, it is necessary to keep it mind that $V_K$ is equivalent to $H_K$, and $R_0$ is equivalent to $H_0$.

## 2.1 STABILITY ANALYSIS

To analyze the dynamical stability of the BAM, we define an energy function $E(t)$ that decreases monotonically over time during inference. The energy function encompasses contributions from both paths: the path from the A end to the B end and the path from the B end to the A end, representing the interaction of states, weights, biases, and their temporal dynamics, as shown in Equation 3.

$$E(t) = -\frac{1}{2} \sum_{k=1}^{K} V_k(t)^T W_k H_{k-1}(t) \tag{3}$$

The time derivative of the energy function can be expressed in Equation 4.

$$\frac{dE(t)}{dt} = -\frac{1}{2} \sum_{k=1}^{K} \left[ \left( \frac{dV_k(t)}{dt} \right)^T W_k H_{k-1}(t) + V_k(t)^T W_k \left( \frac{dH_{k-1}(t)}{dt} \right) \right] \tag{4}$$

Using the dynamics described by the path from the A end to the B end, as shown in Equation 1, and the backward path, as shown in Equation 2, the derivative can be further expanded as shown in Equation 5.

$$\frac{dE(t)}{dt} = -\frac{1}{2} \sum_{k=1}^{K} \left[ \left( \frac{dV_k(t)}{dt} \right)^T \frac{dU_k(t)}{dt} + \left( \frac{dR_{k-1}(t)}{dt} \right)^T \frac{dH_{k-1}(t)}{dt} \right] \tag{5}$$

Then, furthermore, we could obtain the Equation 6.

$$\frac{dE(t)}{dt} = -\frac{1}{2} \sum_{k=1}^{K} \left[ (g \odot (R_k(t))' \left( \frac{dR_k(t)}{dt} \right)^T \frac{dU_k(t)}{dt} + (g \odot (U_{k-1}(t))' \left( \frac{dR_{k-1}(t)}{dt} \right)^T \frac{dU_{k-1}(t)}{dt} \right] \tag{6}$$

Since the activation function $g \odot (\cdot)$ is tanh, sigmoid, or ReLU, it satisfies $\frac{dg}{dt} \geq 0$. Meanwhile, $R_k$ and $U_k$ are at the same layer, and their rates of change have the same sign. Therefore, the inner product of their derivatives is greater than zero. As a result, we have $\frac{dE(t)}{dt} \leq 0$.

This implies that the energy function $E(t)$ does not increase with time, ensuring the dynamical stability of the BAM during inference. The network evolves toward a stable equilibrium, minimizing the energy function during its operation.

## 3 BIDIRECTIONAL SUBSPACE ROTATION ALGORITHM

### 3.1 MATHEMATICAL MECHANISM FOR BIDIRECTIONAL SUBSPACE ROTATION ALGORITHM

In analyzing the mathematical mechanism of BAM, let us start with the most fundamental and original BAM, $Y = sign(WX)$, $X = sign(W^TY)$.

For randomly initialized $\hat{W}$, the outputs are $\hat{Y}$ and $\hat{X}$ respectively. Now, the question becomes how can we rotate the $\hat{W}$ to make the distance between Y and $\hat{Y}$ and X and $\hat{X}$ minimum. Then we need to optimize the Equation 7.

$$\min_{Q^T Q = I_p} \|Y - \hat{Y}Q\|_F + \|X - \hat{X}Q^T\|_F \tag{7}$$

Please refer to Appendix B for details on how this objective is achieved using subspace rotation algorithm.

### 3.2 Pseudo-code for B-SRA and B-BP with two regularizers

In practice, BAM is normally a non-linear system, but the underlying mechanism is the same as described in the Section 3.1. However, we will find the optimization subspace for A and B end alternatively. Then finally, both ends will reach its minimum value. According to the mathematical mechanism mentioned in Section 3.1, we could deduce the B-SRA, as shown in the following pseudo-algorithm 1. While, the B-BP with OWM and GPA regularizers are shown in pseudo-algorithm 2.

---

**Algorithm 1** Bidirectional Subspace Rotation Algorithm

---

**Input**: Samples X, Y, N(Number of Layers) ,and the Epoch
**Output**: weight matrix $W_{ix}$ and $W_{iy}$
**Initialization**: Initialize the orthogonal weight matrices $W_{ix}$ and $W_{iy}$. For convenience, the two ends of the BAM are referred to as follows: the input end $X$ is called $A$, and the input end $Y$ is called $B$. The symbol $\times$ indicates matrix multiplication in this algorithm; we write it explicitly to clearly show the process.
**for** index $\leftarrow$ 1 to Epoch **do**
   Train the BAM from the central weight matrix to the weight matrix at the end
   Alternatively, update the weight matrix at the A end and the weight matrix at the B end
   **for** Counter $\leftarrow$ 1 to N/2 **do**
     Train the Weight Matrix Closed to A End Firstly
     $\hat{A_{ix}}$ = A_FORWARD_B(X, layer=ix)
     $\hat{H_{ix}}$ = B_FORWARD_A(Y, layer=ix)
     $U, \Sigma, V \leftarrow SVD(\hat{A}_{ix}^T \times \hat{H_{ix}})$
     $W_{ix} \leftarrow U \times V \times W_{ix}$
     Train the Weight Matrix Closed to B End Secondly
     iy = N - ix
     $\hat{B_{iy}}$ = B_FORWARD_A(Y, layer=iy)
     $\hat{H_{iy}}$ = A_FORWARD_B(X, layer=iy)
     $U, \Sigma, V \leftarrow SVD(\hat{B}_{iy}^T \times \hat{H_{iy}})$
     $W_{iy} \leftarrow U \times V \times W_{iy}$
   **end for**
   **end for**
   **return** $W_{ix}$ and $W_{iy}$

---

## 4 Experiment and Discussion

### 4.1 Sample Preparation

This paper utilizes patterns from the MNIST dataset, each of which contains 784 nodes ($28 \times 28$), and the character script dataset, which includes regular script ($53 \times 40$) and seal script ($40 \times 40$). The goal of this exploration is to associate paired digit patterns and to associate regular script with its corresponding seal script. All patterns are converted into bipolar form.

---

**Algorithm 2** Bidirectional Back-Propagation with OWM and GPA Regularizes

---

**Input**: Samples X, Y, N(Number of Layers) ,and the Epoch
**Output**: weight matrix $W_i$
**Initialization**: Initialize the orthogonal weight matrices $W_i$. For convenience, the two ends of the BAM are referred to as follows: the input end $X$ is called $A$, and the input end $Y$ is called $B$. The symbol $\times$ indicates matrix multiplication in this algorithm.
**for** index $\leftarrow$ 1 to Epoch **do**
    Train the BAM using the loss value from both A and B end
    While calculating the value of OWM and GPA to regularize the training process
    $\hat{B}$ = A_FORWARD_B(X, layer=N)
    $\hat{A}$ = B_FORWARD_A(Y, layer=0)
    $loss_A$ = MSE($\hat{A}$, A)
    $loss_B$ = MSE($\hat{B}$, B)
    $grad_A$ = GRADIENT($loss_B$, A)
    $grad_B$ = GRADIENT($loss_A$, B)
    $ALIGN$ = CosSim(grad_A, A) + CosSim(grad_B, B)
    $ORTH = \sum_{i=0}^{n}$ MEAN $(W_i \times W_i^T$ - I)
    $LOSS = loss_A + loss_B + \lambda_{align} ALIGN + \lambda_{orth} ORTH$
    Back-propagation the LOSS to Update weight matrix.
**end for**
**return** $W$

---

## 4.2 EXPERIMENT CONFIGURATION

In training BAM, B-BP uses the Adam optimizer with a learning rate set to 0.0001. The output logits from the A end and the B end are used to compute the loss value, and the loss values from both ends are combined to calculate the gradient of the weight matrix. In contrast, when using B-SRA to train BAM, hyperparameters are not required, but the weight matrix is initialized orthogonally. Additionally, for discrete BAM, a sign function is applied to the output during the inference, and the result is compared against the corresponding bipolar pattern to determine whether the BAM can retrieve all patterns correctly iteratively.

## 4.3 EXPLORING THE ROBUSTNESS OF BAM TRAINED BY B-SRA AND B-BP

Figures 1 present initial experiments evaluating the robustness of BAM trained solely with B-SRA and B-BP. In Figure 1(a), we train BAM to associate 20 digit patterns with another 20 digit patterns. When half of the query pattern is masked, the BAM trained by B-BP fails to retrieve the correct associations, demonstrating multiple retrieval errors. In contrast, the BAM trained by B-SRA successfully recalls the target patterns without any error bits, highlighting its larger basin of attraction under partial masking. Similarly, when GN (mean = 0, variance = 1) is added to the query inputs (Figure 1(b)), only the BAM trained by B-SRA retains retrieval accuracy, while the BAM trained by B-BP degrades significantly.

Figure 1(c) and 1(d) further investigates adversarial robustness using FGSM attacks. The BAM trained by B-BP fails completely at even mild perturbation levels ($\epsilon = 0.2$), whereas the BAM trained by B-SRA accurately recalls the target patterns even under strong attacks ($\epsilon = 0.9$). These results consistently show that, in the absence of regularization, the B-BP algorithm fails to produce robust associative memories. In contrast, the BAM trained by B-SRA, by design, produces models that are naturally resilient to noise and adversarial perturbations.

These findings motivate the following section, where we introduce regularization techniques into the B-BP framework. By enforcing weight orthogonality and gradient-pattern alignment, we show that B-BP can be enhanced to achieve or exceed the robustness levels of BAM trained by B-SRA.

### 4.3.1 ROBUSTNESS ANALYSIS FOR BAM TRAINED BY B-SRA

As discussed in Section 4.3, the BAM trained by B-SRA is robust to corrupted or noisy pattern inputs and resilient under adversarial attacks. Through multiple experiments and careful analysis, we

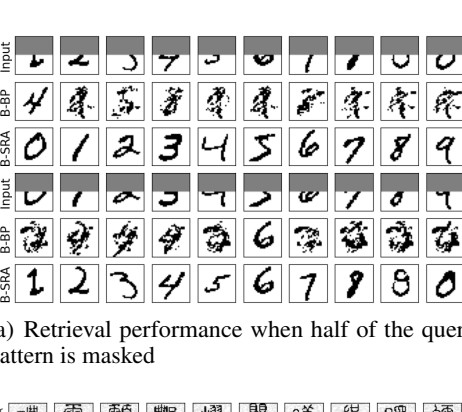
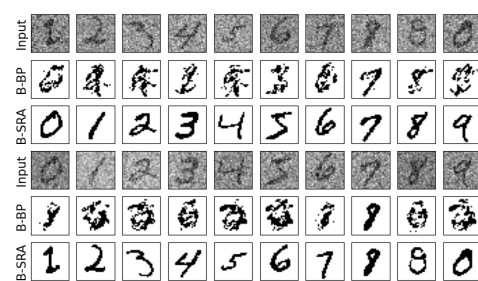

(a) Retrieval performance when half of the query pattern is masked

(b) Retrieval performance under GN perturbation (mean = 0, variance = 1)

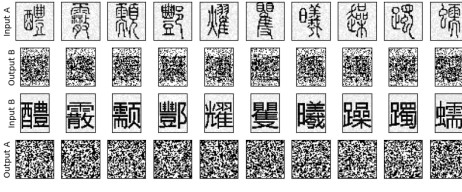
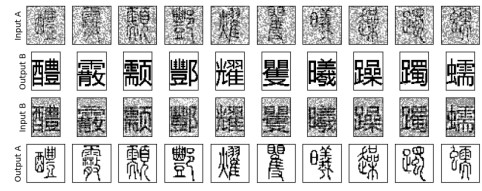

(c) BAM trained by B-BP fails under mild FGSM perturbation ($\epsilon = 0.2$)

(d) BAM trained by B-SRA successfully retrieves patterns under intensive FGSM attack ($\epsilon = 0.9$)

Figure 1: Retrieval performance of BAM trained by B-BP and B-SRA under masking and GN

conclude that two key factors significantly contribute to this robustness: (i) the orthogonality of the weight matrix and (ii) the alignment of the gradient and patterns.

Assuming $f$ is a nonlinear activation function, such as tanh or ReLU, used in the BAM, it typically serves to squash or suppress the magnitude of signals. The robustness provided by the orthogonal matrix can be understood through the lens of a condition number analysis. For an orthogonal matrix, which preserves the norm and has a condition number of 1, the following Equation 8 always holds.

$$\|f[W(x + \delta)\|_F \leq \|W(x + \delta)\|_F = \|Wx + W\delta\|_F = \|x + \delta\|_F \leq \|x\|_F + \|\delta\|_F \tag{8}$$

This ensures that the signal can pass through the network without being distorted, and at the same time, it guarantees that noise is not amplified as the signal propagates from end a to end b, or vice versa.

In terms of the GPA, for a well-trained deep BAM with non-linear activation where the gradient aligns with the patterns, $\frac{\partial \mathcal{L}}{\partial X} = \alpha X$. From the perspective of the loss landscape, it is straightforward to understand that the loss increases most significantly along the direction of X. In other words, the loss landscape is relatively flat in directions perpendicular to X. Typically, noise that is perpendicular to the pattern X is more harmful than noise that is aligned with X, since the latter is inherently suppressed in deep learning models due to activation function damping or cancellation by normalization layers. Therefore, combining GPA with OWM can significantly enhance the robustness of the BAM.

Therefore, to enhance the robustness of BAM trained via B-BP, it is essential to incorporate both the OWM and GPA regularizers into the final objective function. In Section 4.3.2, we analyze how each component contributes to the resilience of BAM.

### 4.3.2 ABLATION STUDY ON REGULARIZATION TECHNIQUES FOR BAM ROBUSTNESS

Table 1: Robustness-Related Metrics: GPA and OWM

| Strategy | GPA(A) | GPA(B) | OWM(A) | OWM(B) |
|---|---|---|---|---|
| SRA | -0.96 ± 0.001 | 0.561 ± 0.001 | 0.0 ± 0.0 | 0.0 ± 0.0 |
| ORTH | -0.31 ± 0.003 | 0.989 ± 0.0 | 18.796 ± 0.705 | 11.159 ± 0.253 |
| SAME | 0.99 ± 0.001 | 0.99 ± 0.0 | 37.442 ± 0.21 | 16.872 ± 0.076 |
| DIFF | -0.979 ± 0.003 | 0.969 ± 0.006 | 10.89 ± 1.0 | 7.268 ± 0.338 |
| ALIGN | 0.99 ± 0.0 | 0.999 ± 0.0 | 695.247 ± 1.997 | 524.527 ± 1.068 |
| BP | -0.09 ± 0.007 | 0.999 ± 0.0 | 684.59 ± 1.236 | 526.135 ± 0.799 |

In this section, we associate 50 pairs of regular and seal script patterns to perform an ablation study and assess the individual contribution of each regularization technique to BAM's robustness. The following abbreviations are used throughout the paper: SRA denotes the Subspace Rotation Algorithm; ORTH refers to BAM trained with the OWM regularizer; SAME applies both OWM and GPA with aligned directions; DIFF uses both regularizers but with opposing alignment; ALIGN applies only GPA without OWM; and BP denotes standard Bidirectional Backpropagation.

Table 2: Robustness of BAM Trained with Different Strategies Under Adversarial Attacks[1]

| Attackers[2] | Strategies | Input A[3] | Output B[3] | Input B[3] | Output A[3] |
|---|---|---|---|---|---|
| | SRA | 12.27 ± 0.058 | 0.674 ± 0.036 | 12.243 ± 0.057 | 0.196 ± 0.036 |
| | ORTH | 12.26 ± 0.057 | **0.06 ± 0.025** | 12.255 ± 0.048 | **0.038 ± 0.002** |
| GN | SAME | 12.217 ± 0.046 | 0.42 ± 0.099 | 12.25 ± 0.036 | 0.066 ± 0.015 |
| | DIFF | 12.279 ± 0.063 | 1.336 ± 0.012 | 12.231 ± 0.05 | 1.311 ± 0.013 |
| | ALIGN | 12.268 ± 0.055 | 1.95 ± 0.007 | 12.251 ± 0.033 | 1.943 ± 0.004 |
| | BP | 12.239 ± 0.052 | 1.964 ± 0.007 | 12.252 ± 0.052 | 1.958 ± 0.006 |
| | SRA | 1.21 ± 0.0 | **0.0 ± 0.0** | 1.21 ± 0.0 | 0.004 ± 0.0 |
| | ORTH | 1.21 ± 0.0 | 0.006 ± 0.003 | 1.21 ± 0.0 | 0.037 ± 0.001 |
| FGSM | SAME | 1.21 ± 0.0 | 0.005 ± 0.004 | 1.21 ± 0.0 | 0.04 ± 0.002 |
| | DIFF | 1.21 ± 0.0 | 0.167 ± 0.012 | 1.21 ± 0.0 | **0.0 ± 0.0** |
| | ALIGN | 1.21 ± 0.0 | 1.867 ± 0.013 | 1.21 ± 0.0 | 1.882 ± 0.007 |
| | BP | 1.21 ± 0.0 | 1.903 ± 0.02 | 1.21 ± 0.0 | 1.899 ± 0.01 |
| | SRA | 1.387 ± 0.004 | **0.0 ± 0.0** | 2.184 ± 0.007 | **0.004 ± 0.0** |
| | ORTH | 1.388 ± 0.004 | 0.004 ± 0.001 | 2.187 ± 0.006 | 0.037 ± 0.001 |
| FFGSM | SAME | 1.388 ± 0.003 | 0.005 ± 0.004 | 2.18 ± 0.004 | 0.041 ± 0.005 |
| | DIFF | 1.387 ± 0.004 | 0.672 ± 0.042 | 2.182 ± 0.006 | 0.985 ± 0.022 |
| | ALIGN | 1.386 ± 0.006 | 1.894 ± 0.008 | 2.183 ± 0.006 | 1.921 ± 0.004 |
| | BP | 1.388 ± 0.005 | 1.924 ± 0.012 | 2.184 ± 0.005 | 1.935 ± 0.014 |
| | SRA | 1.648 ± 0.002 | 0.387 ± 0.051 | 2.042 ± 0.003 | **0.004 ± 0.0** |
| | ORTH | 1.645 ± 0.005 | **0.011 ± 0.006** | 2.042 ± 0.003 | 0.038 ± 0.001 |
| PGD | SAME | 1.645 ± 0.005 | 0.168 ± 0.044 | 2.041 ± 0.004 | 0.04 ± 0.002 |
| | DIFF | 1.649 ± 0.004 | 1.398 ± 0.025 | 2.042 ± 0.004 | 0.581 ± 0.023 |
| | ALIGN | 1.649 ± 0.002 | 1.951 ± 0.008 | 2.041 ± 0.003 | 1.908 ± 0.006 |
| | BP | 1.646 ± 0.006 | 1.964 ± 0.037 | 2.04 ± 0.003 | 1.925 ± 0.006 |

[1] Notes apply to Table 1, 2, 3, and 4.
[2] All attackers of the same type are configured with the same parameters across experiments for fair comparison.
[3] Input A and Input B columns report the mean squared error (MSE) of the adversarial noise added to the respective inputs. Output B and Output A columns show the MSE of the retrieved patterns under perturbation. **Lower output values indicate better robustness**.

As shown in Table 1, the BAM trained by B-SRA achieves optimal values for OWM regularization at both ends, and reasonably good GPA (-0.96 at the a end and 0.561 at the b end). In contrast, while it is challenging for the BAM trained by B-BP to achieve optimal OWM values, it can attain near-optimal GPA values at b end (0.999), even without any regularizers.

It is also observed that the DIFF strategy achieves better OWM values than the SAME strategy (e.g., 10.89 vs. 37.44 at the a end), but subsequent evaluations show that BAM trained with SAME or ORTH demonstrates greater robustness than DIFF, even when the latter has superior OWM metrics, as shown in Figure 2. This discrepancy suggests that negative GPA values, such as the -0.979 seen in DIFF, may contribute to the vulnerability of the BAM under corruption or noise.

Furthermore, the ALIGN strategy achieves nearly perfect GPA at both ends but suffers from extremely high OWM (e.g., over 695 at a end), which significantly reduces its robustness, as shown in Figure 2. These findings indicate that both GPA and OWM are critical and complementary indicators of BAM's robustness. Overemphasis on one while neglecting the other can compromise system reliability.

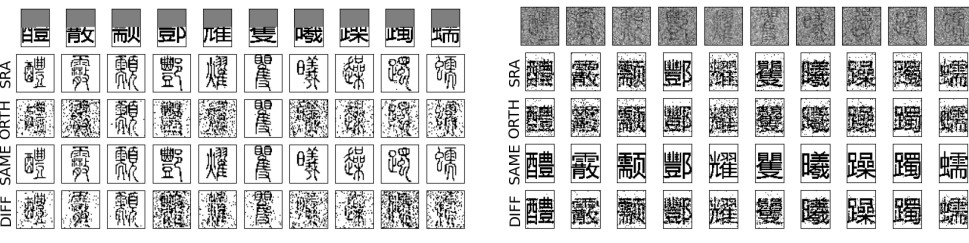

(a) Retrieval performance of models when half of the query pattern is masked

(b) Retrieval performance of models under GN perturbation (mean=0, variance=2)

Figure 2: Comparison of retrieval performance for different models when query patterns are corrupted or noisy

To further evaluate the robustness of BAM trained with different strategies, several adversarial attack approaches are applied, including GN, FGSM, FFGSM, and PGD. As shown in Table 2, under strong GN perturbation, the ORTH strategy performs the best, followed by SAME and then SRA. However, for FGSM and FFGSM attackers, SRA achieves the best performance, while ORTH and SAME perform similarly without significant difference. Under the PGD attacker, SRA is able to retrieve patterns at the b-end with lower error, whereas ORTH and SAME yield better results at the a-end. These results suggest that SRA, ORTH, and SAME strategies are comparably effective in resisting various types of adversarial attacks, each showing strengths under different conditions.

### 4.4 CASE STUDY: BIDIRECTIONALLY ASSOCIATING 100 PAIRS OF REGULAR AND SEAL SCRIPT

(a) Retrieval 100 patterns from corrupted patterns

(b) Retrieval 100 patterns from noisy patterns (mean=0, variance=1.3)

Figure 3: Retrieval performance of BAM trained with different strategies on 100 script pattern pairs

To further evaluate the robustness of BAM trained with different strategies, we conduct an experiment where the BAM is tasked with associating 100 pairs of regular and seal script characters. A quick inspection of Figure 3 shows that when 50% of the input pattern is masked, the SRA and SAME strategies can retrieve the correct patterns with almost no error. Under noisy input conditions (mean = 0, variance = 1.2), the SAME strategy achieves the best performance among all methods. Table 3 further validates these observations by quantifying model robustness under a variety of adversarial attacks. The SAME strategy performs the best under GN, FGSM, and BIM attacks, demonstrating significant advantages in both the A and B directions. For the FFGSM attack, SAME slightly underperforms compared to SRA at the B-end. Under PGD attacks, SAME also performs slightly worse than SRA at the A-end. In conclusion, the SAME strategy consistently performs well across a range of challenging conditions and exhibits the most balanced and reliable robustness profile. These findings suggest that SAME has

strong potential to serve as an optimal training strategy for enhancing the robustness of BAM.

Table 3: Robustness of BAM Trained with Different Strategies on 100 Script Pattern Pairs

| Attackers | Strategies | Input A | Output B | Input B | Output A |
|---|---|---|---|---|---|
| GN | SRA | 1.44 ± 0.004 | 0.407 ± 0.006 | 1.44 ± 0.004 | 0.097 ± 0.007 |
| | ORTH | 1.442 ± 0.004 | 0.557 ± 0.023 | 1.44 ± 0.005 | 0.574 ± 0.04 |
| | SAME | 1.441 ± 0.004 | **0.023 ± 0.014** | 1.44 ± 0.003 | **0.019 ± 0.011** |
| | DIFF | 1.444 ± 0.006 | 0.354 ± 0.013 | 1.44 ± 0.004 | 0.319 ± 0.041 |
| FGSM | SRA | 1.21 ± 0.0 | 2.057 ± 0.008 | 1.21 ± 0.0 | 0.564 ± 0.011 |
| | ORTH | 1.21 ± 0.0 | 1.468 ± 0.065 | 1.21 ± 0.0 | 0.101 ± 0.024 |
| | SAME | 1.21 ± 0.0 | **0.0 ± 0.0** | 1.21 ± 0.0 | **0.0 ± 0.0** |
| | DIFF | 1.21 ± 0.0 | 1.663 ± 0.029 | 1.21 ± 0.0 | 0.0 ± 0.0 |
| FFGSM | SRA | 0.575 ± 0.001 | 0.05 ± 0.003 | 0.95 ± 0.0 | 1.81 ± 0.006 |
| | ORTH | 0.591 ± 0.001 | 0.669 ± 0.047 | 0.93 ± 0.002 | 1.373 ± 0.063 |
| | SAME | 0.58 ± 0.006 | **0.075 ± 0.038** | 0.899 ± 0.007 | **0.115 ± 0.076** |
| | DIFF | 0.57 ± 0.001 | 0.414 ± 0.015 | 0.893 ± 0.001 | 1.774 ± 0.018 |
| BIM | SRA | 0.94 ± 0.001 | 1.842 ± 0.004 | 0.998 ± 0.0 | 0.371 ± 0.009 |
| | ORTH | 0.804 ± 0.004 | 1.937 ± 0.202 | 0.976 ± 0.002 | 0.187 ± 0.042 |
| | SAME | 0.755 ± 0.006 | **0.062 ± 0.066** | 0.975 ± 0.004 | **0.0 ± 0.0** |
| | DIFF | 0.845 ± 0.002 | 1.534 ± 0.017 | 0.947 ± 0.003 | 0.231 ± 0.102 |
| PGD | SRA | 0.885 ± 0.0 | 1.647 ± 0.004 | 0.984 ± 0.0 | **0.113 ± 0.006** |
| | ORTH | 0.822 ± 0.002 | 2.304 ± 0.017 | 0.937 ± 0.002 | 1.055 ± 0.058 |
| | SAME | 0.782 ± 0.006 | **1.258 ± 0.2** | 0.921 ± 0.008 | 0.125 ± 0.062 |
| | DIFF | 0.823 ± 0.002 | 1.383 ± 0.012 | 0.897 ± 0.005 | 0.773 ± 0.058 |

## 4.5 EVALUATING THE RELATIONSHIP BETWEEN CAPACITY AND ROBUSTNESS IN BAM

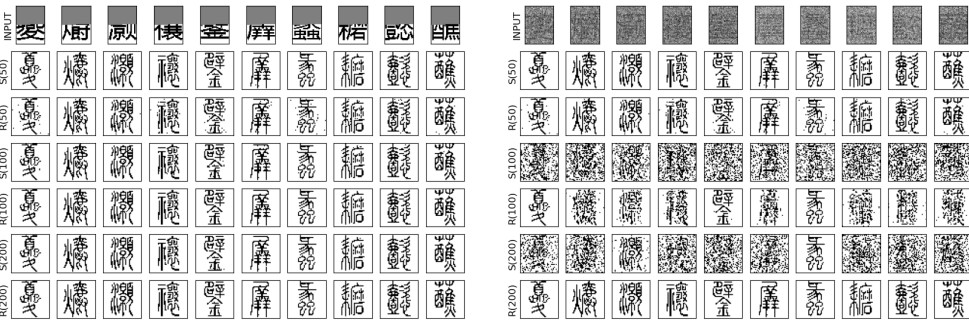

(a) Retrieval performance from corrupted inputs at different capacities

(b) Retrieval performance from noisy inputs at different capacities (mean=0, variance=1.3)

Figure 4: Effect of memory capacity on retrieval performance of BAM trained with different strategies

In this section, we analyze the relationship between memory capacity and the robustness of BAM trained with SRA and SAME by comparing the results presented in Table 4 and Figure 4(a) and 4(b). The BAM models are evaluated under varying memory capacities—associating 50, 100, and 200 pattern pairs—across different adversarial attackers. It is important to note that for storing 50 and 100 pattern pairs, a 3-layer BAM is used, whereas a 5-layer BAM is employed for the 200-pair configuration.

Figure 4(a) shows that, under 50% masking, all models are able to retrieve the correct patterns with relatively low bit errors. However, Figure 4(b) reveals that the BAM trained with SAME to store 200 pairs of patterns (denoted as R(200)) achieves the best performance under noisy input conditions. This indicates that the deeper BAM architecture may contribute to the improved robustness observed under the SAME strategy.

It is observed that for the BAM trained with SRA, retrieval performance gradually degrades with increasing capacity, as shown in Table 4. In contrast, the BAM trained with the SAME strategy to store 200 pairs of patterns achieves the best performance among all models, and the BAM trained with SAME to store 50 pairs performs only slightly worse. This suggests that increasing the number of layers may allow the SAME strategy to fully realize its poten-

tial, yielding the best possible robustness—even as the number of memorized patterns increases.

Table 4: Comparative Study of BAM Robustness Across Memory Sizes (50, 100, 200 Pairs)

| Attackers | Strategies | Input A | Output B | Input B | Output A |
|---|---|---|---|---|---|
| GN | SRA(50) | $2.25 \pm 0.013$ | $0.46 \pm 0.345$ | $2.251 \pm 0.009$ | $0.192 \pm 0.26$ |
| | SAME(50) | $2.253 \pm 0.012$ | $\mathbf{0.057 \pm 0.086}$ | $2.247 \pm 0.008$ | $\mathbf{0.08 \pm 0.098}$ |
| | SRA(100) | $2.247 \pm 0.011$ | $0.455 \pm 0.344$ | $2.252 \pm 0.008$ | $0.195 \pm 0.266$ |
| | SAME(100) | $2.253 \pm 0.01$ | $0.073 \pm 0.1$ | $2.251 \pm 0.008$ | $0.083 \pm 0.098$ |
| | SRA(200) | $2.252 \pm 0.013$ | $0.493 \pm 0.334$ | $2.251 \pm 0.008$ | $0.205 \pm 0.264$ |
| | SAME(200) | $2.253 \pm 0.012$ | $0.061 \pm 0.088$ | $2.246 \pm 0.008$ | $0.082 \pm 0.101$ |
| FGSM | SRA(50) | $1.21 \pm 0.0$ | $1.227 \pm 0.717$ | $1.21 \pm 0.0$ | $0.743 \pm 0.71$ |
| | SAME(50) | $1.21 \pm 0.0$ | $0.002 \pm 0.004$ | $1.21 \pm 0.0$ | $0.013 \pm 0.019$ |
| | SRA(100) | $1.21 \pm 0.0$ | $1.219 \pm 0.737$ | $1.21 \pm 0.0$ | $0.739 \pm 0.711$ |
| | SAME(100) | $1.21 \pm 0.0$ | $0.001 \pm 0.001$ | $1.21 \pm 0.0$ | $0.013 \pm 0.019$ |
| | SRA(200) | $1.21 \pm 0.0$ | $1.293 \pm 0.697$ | $1.21 \pm 0.0$ | $0.796 \pm 0.706$ |
| | SAME(200) | $1.21 \pm 0.0$ | $\mathbf{0.002 \pm 0.004}$ | $1.21 \pm 0.0$ | $\mathbf{0.011 \pm 0.018}$ |
| FFGSM | SRA(50) | $0.565 \pm 0.036$ | $0.02 \pm 0.019$ | $0.962 \pm 0.017$ | $1.913 \pm 0.075$ |
| | SAME(50) | $0.528 \pm 0.036$ | $0.023 \pm 0.03$ | $0.916 \pm 0.017$ | $0.038 \pm 0.043$ |
| | SRA(100) | $0.565 \pm 0.036$ | $0.021 \pm 0.02$ | $0.962 \pm 0.017$ | $1.91 \pm 0.079$ |
| | SAME(100) | $0.528 \pm 0.037$ | $0.034 \pm 0.044$ | $0.916 \pm 0.017$ | $0.056 \pm 0.07$ |
| | SRA(200) | $0.568 \pm 0.035$ | $0.021 \pm 0.019$ | $0.961 \pm 0.017$ | $1.907 \pm 0.074$ |
| | SAME(200) | $0.53 \pm 0.036$ | $\mathbf{0.025 \pm 0.031}$ | $0.915 \pm 0.017$ | $\mathbf{0.037 \pm 0.045}$ |
| BIM | SRA(50) | $0.856 \pm 0.081$ | $1.668 \pm 0.197$ | $0.957 \pm 0.053$ | $0.777 \pm 0.933$ |
| | SAME(50) | $0.652 \pm 0.077$ | $0.694 \pm 0.452$ | $0.875 \pm 0.161$ | $0.255 \pm 0.333$ |
| | SRA(100) | $0.856 \pm 0.082$ | $1.666 \pm 0.201$ | $0.957 \pm 0.053$ | $0.777 \pm 0.936$ |
| | SAME(100) | $0.653 \pm 0.078$ | $0.68 \pm 0.454$ | $0.875 \pm 0.159$ | $\mathbf{0.253 \pm 0.33}$ |
| | SRA(200) | $0.864 \pm 0.078$ | $1.688 \pm 0.189$ | $0.954 \pm 0.054$ | $0.833 \pm 0.942$ |
| | SAME(200) | $0.657 \pm 0.077$ | $\mathbf{0.679 \pm 0.465}$ | $0.866 \pm 0.163$ | $0.27 \pm 0.34$ |
| PGD | SRA(50) | $0.843 \pm 0.045$ | $1.58 \pm 0.116$ | $0.936 \pm 0.06$ | $0.609 \pm 0.819$ |
| | SAME(50) | $0.711 \pm 0.051$ | $\mathbf{1.205 \pm 0.161}$ | $0.859 \pm 0.147$ | $\mathbf{0.29 \pm 0.302}$ |
| | SRA(100) | $0.843 \pm 0.045$ | $1.583 \pm 0.117$ | $0.936 \pm 0.059$ | $0.608 \pm 0.824$ |
| | SAME(100) | $0.71 \pm 0.052$ | $1.228 \pm 0.153$ | $0.859 \pm 0.146$ | $0.297 \pm 0.296$ |
| | SRA(200) | $0.848 \pm 0.043$ | $1.592 \pm 0.112$ | $0.933 \pm 0.061$ | $0.653 \pm 0.831$ |
| | SAME(200) | $0.714 \pm 0.052$ | $1.211 \pm 0.165$ | $0.849 \pm 0.148$ | $0.307 \pm 0.305$ |

## 5 CONCLUSION AND FUTURE STUDY

This paper introduces a novel gradient-free training method, B-SRA, for training BAM. Experimental results show that BAM trained with B-SRA demonstrates strong robustness against adversarial attacks. Motivated by this phenomenon, we identify two key factors that contribute to the robustness of BAM: OWM and GPA. Based on these insights, we design two regularization strategies for B-BP to enhance the resilience of BAM significantly.

Through extensive experiments, including pattern association tasks with digits and Chinese character scripts, we demonstrate that BAM trained with B-SRA achieves superior robustness compared to traditional B-BP. Furthermore, the inclusion of GPA and OWM regularizers in B-BP leads to significant gains in adversarial resilience. Among the training strategies studied, the SAME strategy—using OWM and GPA in the same direction—consistently achieves the best performance, especially in deeper BAM architectures with larger memory capacities.

For future work, we aim to extend our findings from BAM to broader deep learning frameworks. Since BAM shares similarities with the attention mechanism and the architecture of modern Hopfield networks, we plan to incorporate our insights into Transformer and Hopfield-based architectures to develop more robust models. We would also like to develop adversarial attackers specifically designed to target BAM. Since BAM is a purely recurrent neural network, it is fundamentally different from standard feed-forward deep learning models. As such, existing gradient-based attacks may not be suitable for effectively evaluating the vulnerabilities of BAM.

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

TABLE OF CONTENTS

## A    BIDIRECTIONAL BACKPROPAGATION ALGORITHM

B-BP is an extension of traditional backpropagation, designed to optimize both the forward and backward mappings of a neural network (Adigun & Kosko, 2019).

The forward pass maps an input $x_i$ to an output $y_i$ using a function $f(x; \Theta)$, parameterized by $\Theta = \{\theta_0, \theta_1, \cdots, \theta_n\}$. For a dataset of $N$ samples, the forward mapping is represented as in Equation 9.

$$y_i = f(x_i; \Theta), \quad i = 1, 2, \ldots, N \tag{9}$$

The forward error $E_f$ is defined as the sum of losses over all samples in the dataset, as shown in Equation 10.

$$E_f[\Theta] = \frac{1}{N} \sum_{i=1}^{N} \mathcal{L}_f(f(x_i; \Theta), y_i^{\text{true}}) \tag{10}$$

where $\mathcal{L}_f$ is the forward loss function, which can take various forms (e.g., mean squared error, cross-entropy) and $y_i^{\text{true}}$ is the ground truth output for input $x_i$.

The backward pass approximates the reconstruction of the input $x_i$ from the output $y_i$ using the same parameters $\Theta$. The backward mapping is represented as in Equation 11.

$$\hat{x}_i = g(y_i; \Theta), \quad i = 1, 2, \ldots, N \tag{11}$$

The backward error $E_b$ is defined similarly as the sum of losses over all samples in the dataset, as shown in Equation 12.

$$E_b[\Theta] = \frac{1}{N} \sum_{i=1}^{N} \mathcal{L}_b(g(y_i; \Theta), x_i^{\text{true}}) \tag{12}$$

where $\mathcal{L}_b$ is the backward loss function, which can also vary depending on the task. $x_i^{\text{true}}$ is the original input corresponding to the output $y_i$.

The total error to be minimized is the sum of the forward and backward errors, as shown in Equation 13.

$$E[\Theta] = E_f[\Theta] + E_b[\Theta] \tag{13}$$

The gradients of the total error with respect to each parameter $\theta_k$ are calculated, as shown in Equation 14.

$$\Delta\Theta = -\eta \left( \frac{\partial E_f[\Theta]}{\partial \Theta} + \frac{\partial E_b[\Theta]}{\partial \Theta} \right) \tag{14}$$

Parameters $\Theta$ are iteratively updated using these gradients until the neural network converges.

### A.1 REGULARIZATION STRATEGIES FOR ENHANCING B-BP TRAINING

To improve the robustness of the Bidirectional Backpropagation (B-BP) algorithm, we introduce two regularization terms: orthogonal weight matrix (OWM) regularization and gradient-pattern alignment (GPA) regularization. These are applied in the context of a neural network defined abstractly as $Y = f(X)$, with training based on the mean squared error loss:

$$\mathcal{L}_{\text{reconstruction}} = \|\hat{Y} - Y\|^2$$

This regularizer penalizes deviation from orthogonality in the weight matrix $W$, encouraging well-conditioned mappings that preserve input signal magnitudes:

$$\mathcal{L}_{\text{ortho}} = \lambda_{\text{ortho}} \cdot \|W^\top W - I\|_F^2$$

where $\|\cdot\|_F$ is the Frobenius norm, $I$ is the identity matrix, and $\lambda_{\text{ortho}}$ is a coefficient controlling the regularization strength.

This regularizer promotes alignment between the input pattern $X$ and the gradient of the loss with respect to $X$. The alignment is evaluated using cosine similarity:

$$\mathcal{L}_{\text{align}} = \lambda_{\text{align}} \cdot (1 - \cos\theta), \quad \text{where} \quad \cos\theta = \frac{\langle \nabla_X \mathcal{L}, X \rangle}{\|\nabla_X \mathcal{L}\| \cdot \|X\|}$$

Here, $\nabla_X \mathcal{L}$ is the gradient of the loss with respect to input $X$, and $\lambda_{\text{align}}$ balances the contribution of this term.

The full training objective becomes:

$$\mathcal{L}_{\text{total}} = \mathcal{L}_{\text{reconstruction}} + \mathcal{L}_{\text{ortho}} + \mathcal{L}_{\text{align}}$$

# B  MATHEMATICAL MECHANISM FOR BIDIRECTIONAL SUBSPACE ROTATION ALGORITHM

In analyzing the mathematical mechanism of BAM, let us start with the most fundamental and original BAM, which consists of a Hopfield Neural Network (HNN), as shown in Equation 15.

$$\begin{cases} Y = WX \\ X = W^T Y \end{cases} \tag{15}$$

For randomly initialized $\hat{W}$, the outputs are $\hat{Y}$ and $\hat{X}$ respectively. Now, the question becomes how can we rotate the $\hat{W}$ to make the distance between Y and $\hat{Y}$ and X and $\hat{X}$ minimum. Then we need to optimize the Equation 16.

$$\min_{Q^T Q = I_p} \|Y - \hat{Y}Q\|_F + \|X - \hat{X}Q^T\|_F \tag{16}$$

If $Q \in \mathbb{R}^{p \times p}$ is orthogonal, we get Equation 17.

$$\begin{aligned} \|Y - \hat{Y}Q\|_F^2 &= \sum_{k=1}^{p} \|Y(:,k) - \hat{Y}Q(:,k)\|_F^2 \\ &= \sum_{k=1}^{p} (\|Y(:,k)\|_F^2 + \|\hat{Y}Q(:,k)\|_F^2 \\ &\quad - 2Q(:,k)^T \hat{Y}^T Y(:,k)) \\ &= \|Y\|_F^2 + \|\hat{Y}\|_F^2 - 2\sum_{k=1}^{p} [Q^T \hat{Y}^T Y]_{kk} \\ &= \|Y\|_F^2 + \|\hat{Y}\|_F^2 - 2\operatorname{tr}(Q^T (\hat{Y}^T Y)) \end{aligned} \tag{17}$$

Similarly, we could obtain the Equation 18.

$$\|X - \hat{X}Q^T\|_F^2 = \|X\|_F^2 + \|\hat{X}\|_F^2 - 2\operatorname{tr}((\hat{X}^T X Q)) \tag{18}$$

Now, Optimizing the Equation 7 is equivalent to optimizing Equation 19.

$$\max_{Q^T Q = I_p} \operatorname{tr}(Q^T \hat{Y}^T Y) + \operatorname{tr}(\hat{X}^T X Q) \tag{19}$$

It is convenient to observed that the $\operatorname{tr}(Q^T \hat{Y}^T Y)$ and $\operatorname{tr}(\hat{X}^T X Q)$ are equivalent with each other in this case. Then assuming the SVD of $\hat{Y}^T Y$ or $\hat{X}^T X$ are $U^T \Sigma V$ and $V^T \Sigma U$, respectively, then we have Equation 20.

$$\begin{aligned} \operatorname{tr}(Q^T \hat{Y}^T Y) &= \operatorname{tr}(Q^T U^T \Sigma V) = \operatorname{tr}(Q^T U^T V \Sigma) \\ &= \operatorname{tr}(Z\Sigma) = \sum_{i=1}^{p} z_{ii} \sigma_i \leq \sum_{i=1}^{p} \sigma_i \end{aligned} \tag{20}$$

With the same reason, we obtain the Equation 21.

$$\begin{aligned} \operatorname{tr}(\hat{X}^T X Q) &= \operatorname{tr}(V^T \Sigma U Q) = \operatorname{tr}(Q^T U^T V \Sigma) \\ &= \operatorname{tr}(Z\Sigma) = \sum_{i=1}^{p} z_{ii} \sigma_i \leq \sum_{i=1}^{p} \sigma_i \end{aligned} \tag{21}$$

In both equations, $Z$ is an orthogonal matrix defined by $V^T Q^T U$. When $Z$ is an identity matrix, the upper bound is attained, and it is concluded that when $Q = UV^T$, the optimization problem has been solved (Schönemann, 1966).

## C  ADVERSARIAL ATTACK ALGORITHMS

To evaluate the robustness of BAM trained with different strategy, several widely used adversarial attack algorithms, including FGSM, FFGSM, BIM, and PGD, are used. We will briefly discuss each algorithm in this section.

**Fast Gradient Sign Method (FGSM)** (Goodfellow et al., 2014): FGSM is a one-step gradient-based attack that perturbs the input $\mathbf{x}$ in the direction of the gradient of the loss function with respect to the input. The adversarial example is generated as:

$$\mathbf{x}^{\text{adv}} = \mathbf{x} + \epsilon \cdot \text{sign}(\nabla_{\mathbf{x}}\mathcal{L}(\theta, \mathbf{x}, y))$$

where $\epsilon$ controls the perturbation magnitude, and $\mathcal{L}$ is the loss function.

**Fast FGSM (FFGSM)** (Wong et al., 2020): FFGSM is a variant of FGSM that adds random initialization before applying the gradient step to increase attack diversity. It introduces a random perturbation $\delta \sim \text{Uniform}(-\alpha, \alpha)$ to the input before computing the FGSM update.

**Basic Iterative Method (BIM)** (Kurakin et al., 2018): BIM extends FGSM by applying it iteratively with smaller steps. After each step, the perturbation is clipped to ensure it remains within the $\epsilon$-ball around the original input:

$$\mathbf{x}^{\text{adv}}_{t+1} = \text{Clip}_{\mathbf{x},\epsilon}\left\{\mathbf{x}^{\text{adv}}_t + \alpha \cdot \text{sign}(\nabla_{\mathbf{x}}\mathcal{L}(\theta, \mathbf{x}^{\text{adv}}_t, y))\right\}$$

**Projected Gradient Descent (PGD)** (Madry et al., 2017): PGD is a stronger version of BIM with random initialization. It applies iterative updates similar to BIM and projects the adversarial example back onto the allowed $\ell_\infty$-ball centered at the clean input:

$$\mathbf{x}^{\text{adv}}_0 = \mathbf{x} + \delta, \quad \delta \sim \text{Uniform}(-\epsilon, \epsilon)$$

$$\mathbf{x}^{\text{adv}}_{t+1} = \Pi_{\mathcal{B}_\epsilon(\mathbf{x})}\left(\mathbf{x}^{\text{adv}}_t + \alpha \cdot \text{sign}(\nabla_{\mathbf{x}}\mathcal{L}(\theta, \mathbf{x}^{\text{adv}}_t, y))\right)$$

## D  EXTENDED EXPERIMENTAL RESULTS ON BAM ROBUSTNESS

To further support the findings presented in the main text, this appendix provides additional experimental results that examine the robustness and performance of BAM under a broader range of scenarios. These include extended evaluations across multiple datasets, varying memory capacities, and different adversarial conditions. The goal is to reinforce the key observations regarding the effectiveness of the B-SRA algorithm and the proposed regularization strategies when compared to standard B-BP training.

### D.1  INITIAL EXPERIMENT ON B-SRA AND B-BP

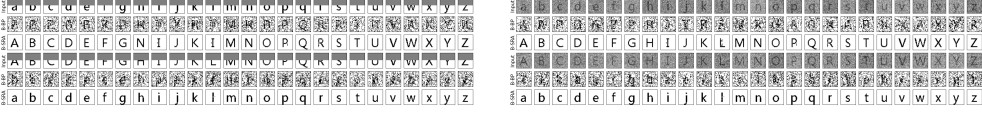

(a) Retrieval performance when half of the query pattern is masked

(b) Retrieval performance under GN perturbation (mean=0, variance=1)

Figure 5: Association of uppercase and lowercase letters using BAM trained with B-BP and B-SRA

To assess the fundamental differences in robustness between B-BP and B-SRA, we conducted a series of initial experiments using three distinct datasets: alphabet letters, MNIST digits, and Chinese script patterns. For each dataset, BAM models were trained using both B-BP and B-SRA, and then evaluated under two adversarial conditions: (i) partially covered patterns (half of the pattern masked), and (ii) Gaussian noise perturbation (mean = 0, variance = 1).

As shown in Figures 5, 6, and 7, the BAM models trained using B-BP consistently failed to recover the correct outputs under both covered and noisy conditions, often not retrieving any effective and

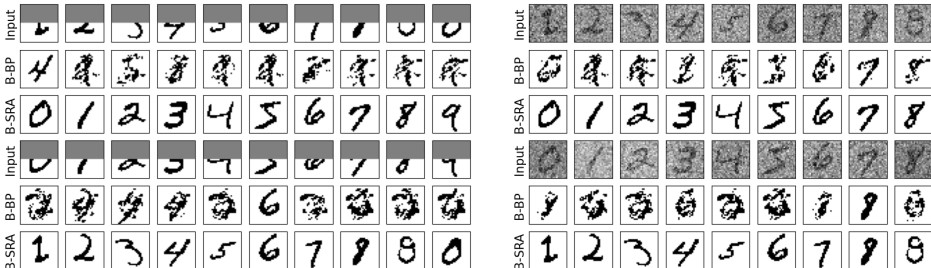

(a) Retrieval performance when half of the query pattern is masked

(b) Retrieval performance under GN perturbation (mean=0, variance=1)

Figure 6: Association of 20 digital number with another 20 digital number in MNIST dataset using BAM trained with B-BP and B-SRA

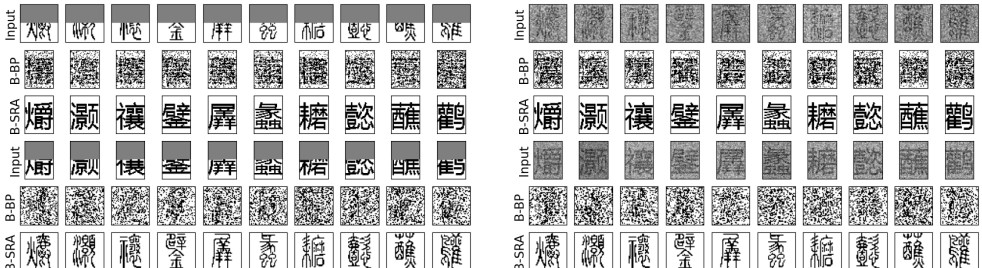

(a) Retrieval performance when half of the query pattern is masked

(b) Retrieval performance under GN perturbation (mean=0, variance=1)

Figure 7: Association of 50 regular scripts with 50 seal scripts using BAM trained with B-BP and B-SRA

clean associated patterns. In contrast, the BAM models trained with B-SRA demonstrated strong resilience, successfully retrieving the clean and clear associated patterns even when the inputs were heavily adversarially perturbed. These results highlight the inherent robustness advantage of B-SRA over B-BP in associative memory tasks.

## D.2 ABLATION STUDY FOR INDIVIDUAL REGULARIZATION

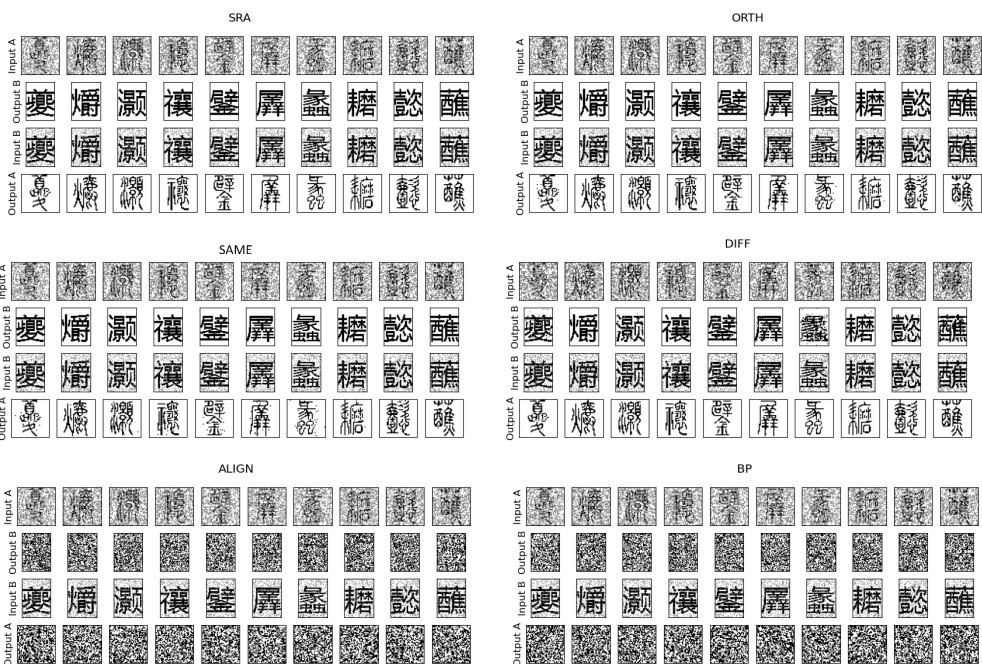

Figure 8: Retrieval performance of BAM trained with different strategies under FGSM attack ($\epsilon = 0.9$)

In this section, we present a comprehensive ablation study using six training strategies—B-BP, ALIGN, SAME, DIFF, ORTH, and SRA—to evaluate the individual contributions of orthogonal weight matrix regularization and gradient-pattern alignment to model robustness. These models are tested under three adversarial attack scenarios: FGSM, FFGSM, and PGD. The corresponding results are visualized in Figures 8, 9, and 10.

It is observed that the models trained with SRA, ORTH, SAME, and DIFF can resist strong FGSM attacks (Figure 8). Under the more aggressive FFGSM and PGD attacks, only SRA, ORTH, and SAME maintain robustness (Figures 9 and 10). Among all configurations, the SAME strategy demonstrates the best overall performance across all attack types.

Notably, for these robust training strategies, the adversarial attacks are unable to generate imperceptible perturbations that deceive the BAM models. To ensure the attack is intensive, we set the attack parameters (e.g., $\alpha$, $\epsilon$) to values significantly larger than those typically used against conventional deep learning models. These findings highlight the inherent robustness of BAM under the SRA, ORTH, and SAME training strategies and suggest the feasibility of embedding BAM modules into broader deep learning frameworks to improve their adversarial resilience.

## D.3 CASE STUDY: ASSOCIATION OF 100 PAIRS OF SCRIPT PATTERNS

To further evaluate the robustness of BAM trained with the SRA, ORTH, SAME, and DIFF strategies, we conducted an additional experiment involving the association of 100 pairs of regular and seal script patterns. The retrieval performance under two adversarial conditions—partially covered patterns and Gaussian noise perturbation—is illustrated in Figures 11 and 12.

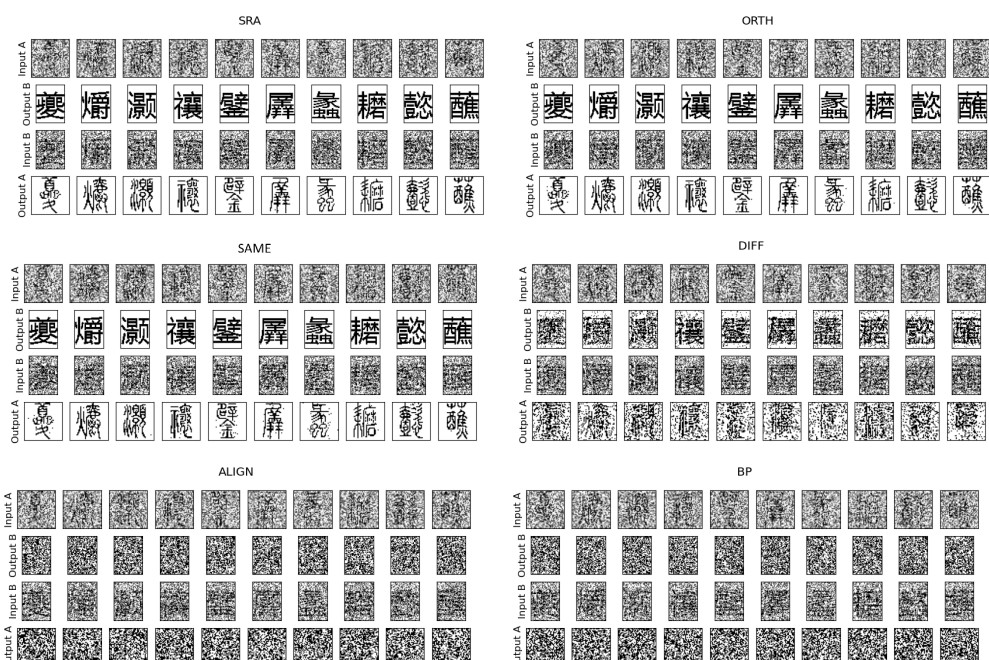

Figure 9: Retrieval performance of BAM trained with different strategies under FFGSM attack ($\alpha = 2$, $\epsilon = 1$)

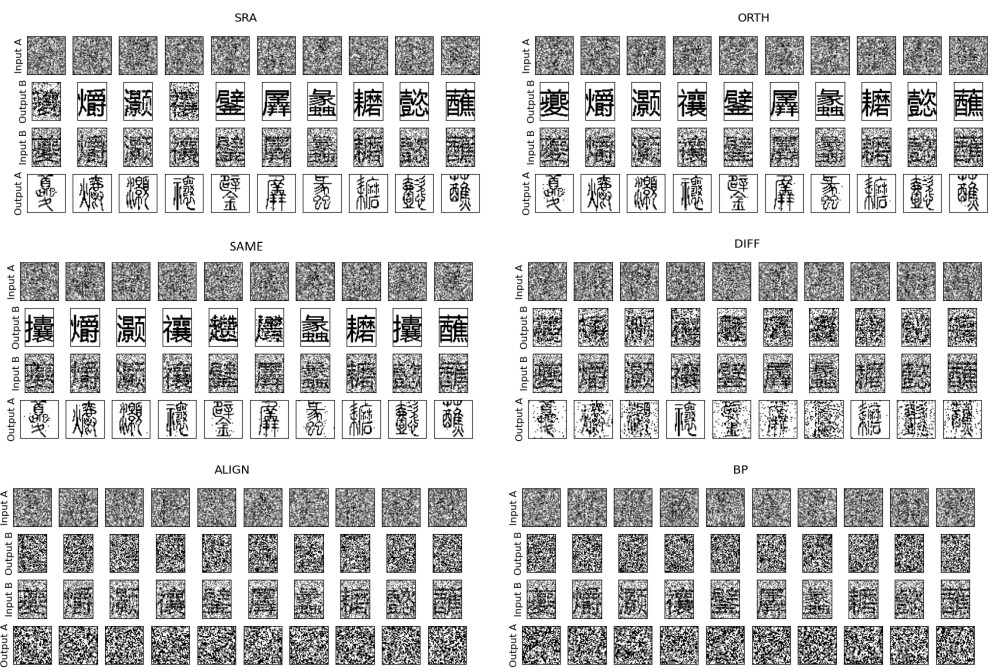

Figure 10: Retrieval performance of BAM trained with different strategies under PGD attack ($\alpha = 2$, $\epsilon = 0.8$, $iteration = 20$)

It is observed that partially covered patterns remain a relatively weak form of attack for all four strategies, with BAM still able to retrieve the associated patterns reliably, as shown in Figure 11. However, when Gaussian noise (mean = 0, variance = 1.2) is added to the inputs, performance differences become more pronounced. In this case, the SAME strategy clearly outperforms both SRA and ORTH, which themselves perform better than DIFF, as shown in Figure 12. These results reaffirm the robustness advantage of SAME, especially under more challenging perturbation scenarios.

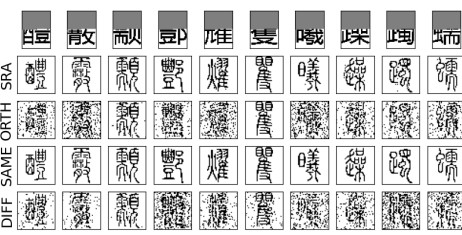
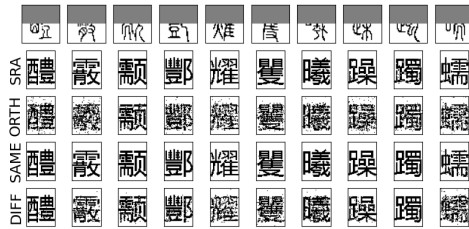

(a) Retrieving 100 patterns from corrupted patterns (from A to B)

(b) Retrieving 100 patterns from corrupted patterns (from B to A)

Figure 11: Retrieving 100 associated patterns from corrupted patterns

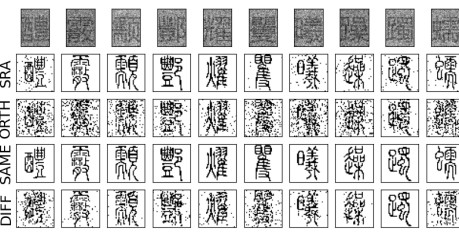
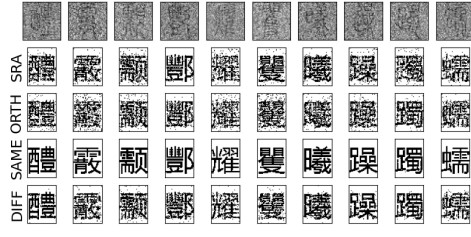

(a) Retrieving 100 patterns from noisy patterns (mean=0, variance=1.2)

(b) Retrieving 100 patterns from noisy patterns (mean=0, variance=1.2)

Figure 12: Retrieving 100 patterns from noisy patterns

Further experiments will be required to comprehensively evaluate the consistency and limitations of these training strategies across more complex datasets and adversarial conditions.

