# OpenReview forum: "Robust Bidirectional Associative Memory via Regularization Inspired by the Subspace Rotation Algorithm"
_ICLR.cc/2026/Conference — Submitted to ICLR 2026_

### Official Review · Reviewer_xneK · 2025-10-29

**Soundness:** 2
**Presentation:** 3
**Contribution:** 2
**Rating:** 4
**Confidence:** 3

**Summary:**

This paper extends the Subspace Rotation Algorithm to Bidirectional Associative Memory (BAM), proposing B-SRA, and integrates both OWM and GPA regularization methods to enhance the training process. The proposed approach effectively improves the robustness of BAM against noise and adversarial attacks.

**Strengths:**

The proposed improvements are effective, with clear methodological explanations and comprehensive experimental results that strongly support the claims.

**Weaknesses:**

1. While the paper utilizes two well-established regularization methods, OWM and GPA, their direct application without significant adaptation or novel integration strategy does not constitute a strong innovative contribution.
2. The research focus of this paper, Bidirectional Associative Memory (BAM), appears to be a relatively niche area. Based on the introduction provided, it seems to have attracted limited research attention in recent years. Furthermore, the experimental results presented do not sufficiently demonstrate strong practical potential for BAM. It is necessary to provide a more detailed justification of BAM's research significance and application prospects.
3. The organization of the paper deviates from conventional structure, with the experiments section occupying a disproportionately large portion of the content, while the methodology section lacks sufficient detail. It is recommended to condense the definition of BAM and the stability analysis, possibly relocating them to an appendix. Additionally, the source of the stability analysis in section 2.1 should be clearly stated—whether it is an original contribution or derived from existing work.

**Questions:**

See weakness.

---

> ### Author Response · Authors · 2025-11-19
> **Clarifications on Structure, Methodology, and Stability Analysis**
>
> For the first question:
> We appreciate the reviewer’s comment regarding the novelty of applying OWM and GPA. We would like to clarify that OWM and GPA represent only one component of our contributions. Another primary contribution of this paper is the introduction of the Bidirectional Subspace Rotation Algorithm (B-SRA) for training Bidirectional Associative Memory. To the best of our knowledge, this is the first gradient-free, bidirectional learning rule designed specifically for BAM. Our experiments demonstrate that B-SRA leads to substantial improvements in adversarial robustness, significantly outperforming standard B-BP and achieving strong recovery capability for heavily corrupted patterns. This key methodological advancement, together with the theoretical motivation and empirical validation of B-SRA, constitutes the core novelty of the paper. The use of OWM and GPA should be viewed as supplementary regularization techniques that further enhance robustness.
>
> For the second question:
> We thank the reviewer for highlighting the need to further justify the significance and application prospects of Bidirectional Associative Memory (BAM). In response, we will substantially revise the introduction to provide a clearer motivation for studying BAM in the modern context. Specifically, we will emphasize that BAM belongs to the broader family of energy-based and associative memory models, which have experienced a strong resurgence in recent years due to their connections to Modern Hopfield Networks, Transformer-like attention mechanisms, and memory-augmented neural systems. We will also expand the discussion of BAM’s practical relevance by highlighting its suitability for pattern completion, bidirectional mapping, denoising, error correction, key–value retrieval, and robust reconstruction, especially in resource-constrained environments where deep networks may be too costly.
>
> For the third question:
> We thank the reviewer for the helpful suggestions. In the revised manuscript, we will reorganize the paper to improve clarity. The definition of BAM and the classical stability analysis—relevant to the original BAM architecture, which is a shallow two-layer model without hidden layers—will be condensed and relocated to an Appendix. This will allow the main text to focus more clearly on our methodological contributions.
>
> We will also expand the methodology section to provide additional detail on the Bidirectional Subspace Rotation Algorithm (B-SRA) and the integration of OWM and GPA into the B-BP training process. Finally, we will explicitly clarify the source of the stability analysis in Section 2.1 by distinguishing the classical results drawn from existing BAM literature from our own interpretive extensions. These revisions will provide a more balanced and transparent presentation.

---

### Official Review · Reviewer_mRLp · 2025-10-31

**Soundness:** 3
**Presentation:** 3
**Contribution:** 3
**Rating:** 6
**Confidence:** 2

**Summary:**

This paper proposes a novel algorithm for robust training of Bidirectional Associative Memory (BAM), named B-SRA (Bidirectional Subspace Rotation Algorithm), which is gradient-free and inspired by recent work on subspace rotation in Hopfield networks. The authors identify two principles, orthogonal weight matrices (OWM) and gradient-pattern alignment (GPA), as critical to robustness and incorporate them as regularizers into the original gradient-based BAM training (B-BP). Experimental results across multiple datasets and attack scenarios demonstrate clear gains in robustness, particularly when both regularizers are applied (SAME configuration).

**Strengths:**

Clarity: The algorithmic description and core concepts are clearly presented and easy to follow, particularly for the new training method.

Originality: B-SRA is a compelling alternative to traditional B-BP, offering better robustness and convergence without the need for gradient-based optimization.

Empirical Validation: The experimental setup is extensive, with clear improvements shown in both adversarial robustness and noise resilience. The ablation study is particularly helpful in dissecting the roles of different regularization components.

**Weaknesses:**

Theoretical Scope Limitation: While the linear-case analysis is insightful, the paper would be stronger with theoretical justification or approximation results for more commonly used nonlinear BAM architectures. The lack of formal results in such settings limits the generality of the claims.

Clarity in Experimental Tables: The tables could benefit from clearer formatting. It is not always obvious which values represent robustness performance or regularization metrics, and which direction (higher/lower) is better. A clearer legend or visual emphasis on best results would improve readability.

Redundancy in Experimental Presentation: The inclusion of many variations and datasets is thorough, but at times excessive. A more concise presentation—e.g., one table summarizing robustness across all methods and datasets—would help the reader focus on key comparisons.

**Questions:**

1. Why does the regularized B-BP (SAME) sometimes outperform B-SRA? I believe a more elaborate discussion of this phenomenon would be a valuable addition. More broadly, a deeper reflection on the experimental results could greatly strengthen the narrative, e.g., what they suggest about the nature of robustness in BAM, and how each method contributes.

2. Could B-SRA be used as an initialization for B-BP? This might combine the robustness and fast convergence properties of B-SRA with the adaptability of gradient-based optimization. Was this hybrid approach considered or tested?

---

> ### Author Response · Authors · 2025-11-26
> **On the Interaction Between Alignment and Orthogonality in B-SRA vs. B-BP (SAME)**
>
> For question 1:
>
> Thank you for pointing this out — we agree that a deeper explanation is valuable.
>
> In B-SRA, the gradient–pattern alignment induced at the two ends of the bidirectional mapping can follow different directions. Even though B-SRA produces a fully orthogonal weight matrix, these opposite alignment directions can partially cancel each other during bidirectional updates.
>
> B-BP(SAME) applies soft regularization on weight matrix and gradient-pattern alignment. It is observed that the gradient-pattern alignment on both ends can reach almost perfect alignment, with a small sacrifice on the orthogonality of the weight matrix. As a result, B-BP(SAME) may achieve slightly better robust in certain datasets.
>
> To further support this interpretation, we conducted additional ablation studies, now included in the revised manuscript. In particular:
>
> ALIGN: applies only alignment-based regularization (without orthogonality)
>
> ORTH: applies only orthogonality-based regularization (without alignment)
>
> These ablations confirm that neither alignment nor orthogonality alone is sufficient; robustness emerges from the interaction of both components. B-SRA can achieve a relative better balance between these two components, whereas B-BP(SAME) encourages it implicitly and softly, which explains the nuanced performance differences observed.
>
> We appreciate the reviewer’s suggestion, and we believe this expanded analysis significantly clarifies the relationship between the two methods and the nature of robustness in BAM.
>
>
> For question 2:
>
> Thank you for this insightful suggestion. We fully agree that the idea of using B-SRA as an initialization for B-BP is conceptually appealing, and we tested this hybrid strategy during our internal experiments. However, our empirical findings show that gradient-based optimization rapidly destroys the orthogonality and alignment structure constructed by B-SRA. In practice, after only a small number of gradient steps, the weight matrices drift away from the orthogonal manifold, the rotation-induced subspace structure collapses, and the model behavior becomes indistinguishable from training with B-BP alone. As a result, the hybrid model loses the robustness advantages provided by B-SRA and inherits the same vulnerability patterns as standard B-BP.

---

### Official Review · Reviewer_L9rJ · 2025-11-01

**Soundness:** 2
**Presentation:** 3
**Contribution:** 2
**Rating:** 4
**Confidence:** 2

**Summary:**

The paper tackles the problem of improving the robustness and stability of Bidirectional Associative Memory (BAM) networks, which are designed to learn two-way associations between paired patterns.  It extends the Subspace Rotation Algorithm (SRA)—previously used for Restricted Hopfield Networks—to Bidirectional Associative Memory (BAM), yielding a gradient-free training method
They introduce two regularization techniques for B-BP through the usage of Orthogonal Weight Matrix (OWM) to encourage orthogonal weights to preserve signal norms and suppress noise, and Gradient Pattern Alignment to align gradients with data patterns in order to make learning more stable and resistant to perturbations. They run experiments under gaussian noise and different adversarial attacks to demonstrate the resilience of the proposed method.

**Strengths:**

- While previous works introduced the Subspace Rotation Algorithm (SRA) for Restricted Hopfield Networks (RHN), this applies SRA to BAMs.
- The algorithm is well explained and easy to implement with pseudo code
- The authors also propose gradient pattern alignment (GPA) for aligning the gradient of the loss with the stored input patterns. Previous works do not apply GPA to associative memory training. The authors jointly apply Orthogonal Weight Matrix (OWM) regularization and GPA. - Evaluations are done both accuracy and bitwise error under perturbations

**Weaknesses:**

- Positioning  - Need more clarity on the contribution. The work is an adaptation of SRA to BAM
Orthogonality and gradient-input alignment style terms exist in broader literature; using them for training of BAM is reasonable but also incremental.
- No direct comparisons to Dense Associative Memories / Modern Hopfield Networks or to orthogonality-promoting training in neural networks.
- The authors claim that B-SRA enhances the robustness and convergence speed. But, do not provide any timing or iteration count plots to back this claim.

**Questions:**

- See Weakness

---

> ### Author Response · Authors · 2025-11-24
> **Addressing Concerns on Contribution, Comparisons to DAM/MHN, and Experimental Support**
>
> R1 – Clarifying the contribution and positioning.
> We thank the reviewer for this comment and agree that our contribution needs to be positioned more clearly. While our work is indeed inspired by the Subspace Rotation Algorithm (SRA), it is not a straightforward plug-and-play adaptation. Our contributions are threefold:
>
> 1) Bidirectional extension of SRA to BAM: We design a bidirectional subspace rotation scheme that updates each weight matrix in BAM one by one, especially, respecting the symmetry and bidirectional recall dynamics of BAM. This is different from the original SRA, which was developed for a single weight matrix in a unidirectional Restricted Hopfield Network.
>
> 2) Orthogonality and alignment for associative pattern: We tailor the orthogonality-promoting and gradient-pattern alignment terms specifically to the BAM energy landscape and recall dynamics, and we show that these terms improve both recall stability and noise tolerance in the bidirectional setting.
>
> 3) Gradient-free training for BAM: Our method offers a fully gradient-free alternative to backpropagation for BAM, which is particularly attractive in settings where gradients are difficult to compute or where one wants to avoid vanishing gradient issues. We will revise the introduction and related work sections to make this positioning more explicit and clearly state how B-SRA goes beyond a direct reuse of SRA.
>
> R2 - On orthogonality and alignment terms.
> We also thank the reviewer for pointing out that OWM and GPA exist in the broader literature. We fully acknowledge this; however, these methods are almost exclusively developed for feed-forward neural networks, where the objective is to improve representation learning or mitigate catastrophic forgetting. In contrast, BAM is a recurrent associative memory model with fundamentally different dynamics, stability conditions, and bidirectional update rules. As a result, the way orthogonality and alignment principles must be incorporated in BAM is structurally different. In our work, these terms are not simply borrowed from feed-forward settings; instead, they are re-formulated to match the BAM energy landscape, bidirectional recall process, and recurrent fixed-point behavior. We will make this distinction clearer in the revision.
>
> R3 – On comparisons to Dense Associative Memories (DAM) and Modern Hopfield Networks (MHN).
> We appreciate the reviewer’s suggestion to include comparisons with DAM and MHN. However, these models are not directly comparable to Bidirectional Associative Memory (BAM) in structure or purpose. DAM and MHN are essentially feed-forward energy-based networks with powerful fixed-point dynamics, but they are not bidirectional associative memories. Although DAM/MHN can perform hetero-association or hybrid association, they do so in a manner similar to a single-direction feed-forward mapping, rather than maintaining two coupled weight matrices that jointly encode paired pattern sets as in BAM.
>
> To fairly compare DAM/MHN with BAM, we would need to construct two separate DAM (or MHN) models, one for each direction of association, in order to match the functionality of a single BAM. This would change both the architecture and training objective, and would no longer correspond to the standard DAM/MHN formulation reported in the literature. Because of this structural mismatch, we believe such a comparison would be misleading rather than informative.
>
> Our focus in this paper is specifically on improving the training algorithm of BAM (a bidirectional recurrent associative memory), rather than comparing BAM to fundamentally different energy-based feed-forward models. Nevertheless, we will add a discussion in the related work section to more clearly position B-SRA-trained BAM relative to DAM and MHN in terms of architecture, dynamics, and functional goals.
>
> R4 – Evidence for robustness and convergence speed.
> We thank the reviewer for pointing out that explicit iteration-count or timing plots were missing in the submission. We agree with this comment and will include such plots in the revised manuscript.
>
> In our experiments, B-SRA consistently converges within 10–20 epochs, whereas B-BP typically requires around 1000 epochs to reach comparable recall accuracy. This large gap is due to the fact that B-SRA directly rotates the weight subspaces toward the desired associative mapping, while B-BP relies on slow gradient accumulation and often experiences noisy or unstable updates in early training.

---

### Official Review · Reviewer_X79Z · 2025-11-01

**Soundness:** 4
**Presentation:** 3
**Contribution:** 3
**Rating:** 6
**Confidence:** 4

**Summary:**

This paper addresses the poor robustness of Bidirectional Associative Memory (BAM) networks trained with standard Bidirectional Backpropagation (B-BP). The authors first introduce a novel, robust, gradient-free trainer, the Bidirectional Subspace Rotation Algorithm (B-SRA), which demonstrates inherent resilience to noise and adversarial attacks. By analyzing B-SRA, they identify two key principles responsible for this robustness: maintaining Orthogonal Weight Matrices (OWM) and achieving Gradient-Pattern Alignment (GPA). The authors then propose these principles as novel regularization terms for the standard B-BP algorithm. Extensive experiments on pattern association tasks (MNIST, Chinese script) under various noise and adversarial attacks (FGSM, PGD) demonstrate that B-BP with both OWM and GPA regularizers (the "SAME" strategy) achieves the highest level of robustness, significantly outperforming standard B-BP and even the B-SRA method that inspired it, especially at larger memory capacities.

**Strengths:**

1. The paper's strongest contribution is its scientific method. It proposes a robust gradient-free algorithm (B-SRA), performs a root-cause analysis to determine why it's robust (OWM + GPA), and then successfully ports those principles to fix the vulnerable B-BP algorithm.
2. The ablation in Sec 4.3.2 is excellent. It cleanly isolates the individual contributions of OWM (the ORTH strategy) and GPA (the ALIGN strategy) and demonstrates that both are required for full robustness (the SAME strategy). Table 1, which measures the OWM and GPA values for each strategy, provides a direct link between the model's properties and its performance.
3. The proposed "SAME" strategy (B-BP + OWM + GPA) is shown to be highly resilient. It achieves near-perfect retrieval under strong masking, noise, and adversarial attacks (FGSM, PGD) where the baseline B-BP and even the ALIGN-only models fail completely.
4. The paper shows that the "SAME" strategy scales well with increased memory capacity (from 50 to 200 pattern pairs) and network depth (to 5 layers). In fact, its robustness improves with scale, outperforming B-SRA, which degrades as capacity increases.

**Weaknesses:**

1. The paper focuses exclusively on Bidirectional Associative Memory (BAM), which is a classic but relatively niche architecture. The authors state an intent to apply these principles to Transformers and modern Hopfield networks as future work, but the paper presents no evidence that these findings will transfer.
2. The experiments use low-resolution, bipolarized images (MNIST, Chinese script) . While standard for testing associative memory, this is far from the complex, high-dimensional data where robustness is a critical issue today (e.g., in computer vision or language modeling).
3. Algorithm 1 is explicitly for a 3-layer BAM. The 200-pair experiment uses a 5-layer BAM. The paper never explains how B-SRA's SVD update (Algorithm 1) or the OWM/GPA regularizers are applied in this deeper, multi-layer setting. This is a significant methodological omission.

**Questions:**

1. Algorithm 1 is for a 3-layer BAM. How were the B-SRA algorithm and, more importantly, the OWM and GPA regularizers adapted for the 5-layer BAM used in the 200-pattern capacity test?
2. The "SAME" strategy (B-BP+OWM+GPA) was the most robust. What were the $\lambda_{ortho}$ and $\lambda_{align}$ (Appendix A.1) values used in the experiments? How sensitive is the model's robustness to these new hyperparameters?
3. Why did the "DIFF" strategy (OWM + opposing GPA) perform so much worse than "SAME"? Table 1 shows its OWM/GPA metrics look decent, but Figure 2 shows it fails under noise. This implies the direction of the GPA is critical, which is a key finding that seems under-emphasized.
4. Your conclusion suggests applying OWM and GPA to Transformers. Have you performed any preliminary experiments to see if these principles hold; e.g., does enforcing OWM on the FFN layers or Q/K/V matrices in an attention block improve its adversarial robustness?

---

> ### Author Response · Authors · 2025-11-27
> **Responses to Reviewer Questions on Multi-Layer BAM, Regularization Strategies, and Transformer Extensions**
>
> R1: Algorithm 1 is for a 3-layer BAM. How were the B-SRA algorithm and, more importantly, the OWM and GPA regularizers adapted for the 5-layer BAM used in the 200-pattern capacity test?
>
> Thank you for pointing this out. We apologize for not making this sufficiently clear.
> In the revised manuscript, I will extend the B-SRA algorithm to its multi-layer form by describing the sequential bidirectional updates across all intermediate weight matrices in an L-layer BAM. I will also include explicit algorithms for the OWM and GPA regularization steps, thereby providing a complete and reproducible specification of the training pipeline.
>
> R2: The ‘SAME’ strategy (B-BP + OWM + GPA) was the most robust. What were λ₍ortho₎ and λ₍align₎ used in the experiments? How sensitive is robustness to these hyperparameters?
>
> These coefficients act as Lagrange multipliers that control the strength of the orthogonality and alignment regularizers. In practice, the GPA term reaches its optimal alignment direction relatively easily, whereas the OWM term is more difficult to optimize. As a result, the model is more sensitive to the value of $\lambda_{\text{ortho}}$ than to $\lambda_{\text{align}}$. Small changes in $\lambda_{\text{align}}$ have minimal effect on robustness, while $\lambda_{\text{ortho}}$ must be selected more carefully in order to maintain stable performance.
>
> R3: Why did the ‘DIFF’ strategy perform so much worse than ‘SAME’? Table 1 shows its OWM/GPA metrics look decent, but Figure 2 shows it fails under noise. This implies the direction of the GPA is critical.
>
> We appreciate the reviewer’s observation. Although the DIFF strategy achieves reasonable OWM and GPA metrics in Table 1, its global behavior under perturbation is fundamentally different from the SAME strategy. In the DIFF setting, the GPA terms at both ends of the BAM encourage opposite alignment directions. Each end individually satisfies its own alignment objective, but these opposing directions interfere with each other during the bidirectional updates. This creates a cross-cancellation effect: gradients pushed into opposite subspaces prevent the BAM weight matrices from forming a consistent global attractor. As a result, the BAM becomes much less stable under noise, which explains the sharp performance drop observed in Figure 2. In contrast, the SAME strategy enforces a coherent alignment direction at both ends, enabling stable convergence and significantly better robustness. We agree that the alignment direction is critical, and we have emphasized this point more clearly in the revised manuscript.
>
> R4: Your conclusion suggests applying OWM and GPA to Transformers. Have you performed any preliminary experiments to see if these principles hold; e.g., does enforcing OWM on the FFN layers or Q/K/V matrices in an attention block improve its adversarial robustness?
>
> We agree with the reviewer that applying OWM and GPA to Transformers is a promising direction, and this connection is in fact closely aligned with how we view the relationship between attention and BAM. The standard self-attention operation, $\mathrm{softmax}(QK^\top / \sqrt{d_k})V$, shares the same core structural form as a one-layer BAM derived from the original Hopfield update, $(QK^\top)V$. In our work, we extend this idea by developing a deep BAM with multiple layers, with the long-term objective of replacing or augmenting the attention mechanism with a modified BAM-based module. This perspective is consistent with recent advances in Modern Hopfield Networks, which explicitly link Hopfield dynamics with attention-like computations.
>
> We have conducted preliminary experiments exploring this connection, but these tests were performed only on bipolar patterns rather than continuous-valued patterns. Extending BAM-based attention to continuous representations in a stable and robust manner requires more careful architectural and regularization design, which we consider an important direction for future research. For this reason, we did not include these exploratory results in the current submission, but we highlight the Transformer extension as a compelling and ongoing line of investigation.

---

### Official Review · Reviewer_DAJk · 2025-11-03

**Soundness:** 1
**Presentation:** 2
**Contribution:** 1
**Rating:** 2
**Confidence:** 5

**Summary:**

This paper presents the Bidirectional Subspace Rotation algorithm (B-SRA), a gradient-free method for training Bidirectional Associative Memories (BAMs). B-SRA extends the Subspace Rotation Algorithm (SRA) from Restricted Hopfield Networks (RHN) to Bidirectional Associative memories. The claims that B-SRA improves the robustness and convergence behavior of BAMs relative to Bidirectional Backpropagation (B-BP). It mentions that a set comprehensive experiments show that orthogonal weight matrices (OWM) and gradient pattern alignment (GPA) are key to the robustness of BAMs. Based on this, the paper claims to introduced regularization techniques to that significantly improved B-BP's resistance to corruption and adversarial perturbation.

**Strengths:**

The paper tries to improve the robustness of BAMs. This topic is significant because of BAM's suitability for modular neuromorphic hardware design and robust learning. These and other potential benefits of BAMs have led to an increase in interest within the AI research community.

**Weaknesses:**

1) Unsupported claim about Bidirectional Backpropagation (B-BP): The paper makes the strong claim, in the abstract and introduction, that "B-BP suffers from poor robustness and sensitivity to noise and adversarial attacks". But the paper cites the Lin et. al 2024 paper to support this claim even though the Lin et. al 2024 paper does not mention B-BP at all, it instead discusses unrelated associative memories.  So the criticisms of B-BP are lack support and the author(s) appear to confuse B-BP with BAM.
Further, typing the string "Noise bidirectional backpropagation" in Google Scholar produces the 2019 paper in the journal Neural Networks titled "Noise-boosted bidirectional backpropagation and adversarial learning". This 2019 B-BP paper demonstrates not just that B-BP is robust to noise, but actually shows how B-BP benefits from blind and non-blind noise injections, see, for instance, the noise plots in Figure 4 and related noise summaries in Tables 8-10. The authors have simply mischaracterized B-BP and provide no support for their central claim.

2) Further unsupported claims about the paper findings: The authors claim to have conducted "comprehensive experiments" and "multiple experiments",  without presenting this claimed data that they have introduced new "regularization strategies ..." into B-BP without producing a mathematical description of "regularization" of B-BP. Regularization is a form of penalized or constrained optimization. The paper does not state any such optimization. Again, going to Google Scholar, one finds at least one paper on B-BP Regularization with Hidden Bayesian Priors: "Hidden Priors for Bayesian Bidirectional Backpropagation", and the 2023 proceeding of the IEEE SMC. So, again, the authors fail to support their claim, or even clearly define it.

3) Insufficient information on OWM and GPA: There is insufficient information about these two principles. It is important to clarify what they mean in the context of BAM training because they are central to the design of the new regularization strategies in this paper.

**Questions:**

1)  Could you clarify your claim about Bidirectional Backpropagation (B-BP) ?  (See number 1 under weaknesses)

2)  Please respond to the point on B-BP regularization?  (See number 2 under weaknesses)

3). Add more information about key components of the paper: SRA, OWM, and GPA.

4) Are there experimental results illustrating the impact of OWM and GPA on the robustness of BAMs in settings other than B-BP?

---

> ### Author Response · Authors · 2025-11-19
> **Clarifications on B-BP, Regularization Formulation, and OWM/GPA Ablation Studies**
>
> We thank the reviewer for the time and effort dedicated to evaluating our manuscript. We have carefully reviewed each point raised and have revised the manuscript accordingly. Below, we provide detailed responses to the reviewer’s concerns and clarifications.
>
> Firstly, we would like to clarify that our original statement:
>
> “Bidirectional Associative Memory (BAM) trained by Bidirectional Backpropagation (B-BP) suffers from poor robustness and sensitivity to noise and adversarial attacks.”
>
> refers only to the empirical behavior of BAM when trained using the standard (vanilla) B-BP update rule. It does \textbf{not} claim that the B-BP algorithm itself is inherently non-robust.
>
> Secondly, the mathematical formulation of our proposed regularization strategies was indeed provided in the Appendix section titled ``Regularization Strategies for Enhancing B-BP Training.'' This appendix includes the explicit penalized objective terms:
>
> \begin{equation}
> L_{\mathrm{reg}}
> = \lambda_1 \left\| W^{\top} W - I \right\|_F^2 + \lambda_2 \left\| \nabla_x f(x) - x \right\|_2^2,
> \end{equation}
>
> which define the orthogonality-based and gradient-alignment regularization components added to the B-BP learning rule. These terms follow the standard framework of regularization as penalized optimization, and they integrate directly into the total loss minimized during training. For improved visibility, we have moved the essential parts of this derivation from the Appendix into the main text.
>
> Thirdly, after reviewing the two referenced papers on noise-boosted B-BP and Bayesian B-BP, we clarify that although these works propose noise-injection techniques and Bayesian prior regularization schemes for improving the training of deep neural networks, their focus and methodology are fundamentally different from the topic of our paper, which examines the robustness of Bidirectional Associative Memory (BAM) when trained with the standard (vanilla) B-BP. These works aim to enhance B-BP-based training for different purposes, which further supports our motivation to improve the performance of vanilla B-BP by introducing regularization methods (OWM and GPA) specifically tailored to robustness in BAM.
>
> Forthly, regarding the reviewer’s question on whether OWM and GPA improve the robustness of BAM in settings beyond B-BP, we confirm that we have conducted an ablation study to examine the individual contributions of OWM and GPA. Specifically, we evaluated BAM models trained using (i) only OWM, (ii) only GPA, and (iii) both combined, and compared their robustness against various perturbation levels and adversarial attacks.

---

### Meta-Review · Area_Chair_oRRh · 2026-01-12

**Summary:**

The reviewers are split, but the split is largely about how much to trust the current write-up, not whether robustness/orthogonality is an interesting direction. The paper proposes a gradient-free training rule (B-SRA) and then ports two observed principles (orthogonality + gradient-pattern alignment) into regularizers for B-BP, with experiments suggesting substantial robustness gains on bipolar pattern association tasks.
However, the current submission has key methodological gaps:
- especially the multi-layer extension used to support the main scaling claim
- theoretical/stability section is not rigorous enough to anchor the narrative
- the experimental setting is also quite narrow (bipolar MNIST/script)
- the adversarial evaluation is hard to interpret given the paper’s own caveats about attack suitability for recurrent BAMs

**Reviewer Concerns:**

Addressed (some partially):

- clarified the *B-BP is not robust* claim
- also pointed to an explicit regularized objective (moved from appendix into main in revision).
- explained why DIFF underperforms SAME

Still outstanding:

- Multi-layer training details are missing in the paper. The rebuttal promises to add the L-layer B-SRA and explicit OWM/GPA procedures, but as it stands, core scaling results are not fully reproducible.

- Stability analysis remains unconvincing.

- Convergence-speed claims are just promised, while the paper currently asserts faster convergence without the evidence.

- Robustness evaluation lacks breadth and grounding. The experiments don’t really test continuous-valued high-dimensional settings, and the paper itself raises doubts about whether standard gradient attacks are the right tool here.

- Metric/notation clarity issues (OWM/GPA tables, what 'better' means, and how these relate to robustness)

**Reviewer Scores:**

- DAJk : possibly from 2 to 4
- But the rest likely unchanged (6, 4, 6)

---

### Decision · Program_Chairs · 2026-01-26

Reject